# From Agricultural Green Production to Farmers’ Happiness: A Case Study of Kiwi Growers in China

**DOI:** 10.3390/ijerph20042856

**Published:** 2023-02-06

**Authors:** Wen Xiang, Jianzhong Gao

**Affiliations:** College of Economics and Management, Northwest A&F University, Taicheng Road, Yangling 712100, China

**Keywords:** agricultural green production, happiness, farmers, Shanxi, northwest China

## Abstract

Happiness is the ultimate goal of everyone working and earning wealth. At present, in the vast rural areas of China, the excessive and unscientific application of chemical fertilizers and pesticides is raising serious concerns for environmental protection. The Chinese government has strongly promoted agricultural green production as an alternative to the previous agricultural paradigm that sacrificed the environment. A shift towards agricultural green production has become imperative. However, will it bring “happiness” to farmers who partake in this shift? Using data from 1138 farmers in Shanxi, Northwest China, throughout 2022, this article examines the relationship between the adoption of agricultural green production and farmers’ happiness. The empirical findings indicate that the adoption of agricultural green production can considerably increase farmers’ happiness and that the more types of agricultural green technologies that are implemented, the greater the happiness of farmers. Further mediating effect analysis suggests that this mechanism occurs through raising the absolute and relative income, mitigating agricultural pollution, and elevating the social status. The findings shed new light on the effect of farmers’ economic behavior on their happiness and underline the necessity of implementing relevant policies.

## 1. Introduction

Since the start of reform and the opening up of the economy, agriculture has played a prominent role in China’s remarkable economic success [1]. However, economic development at the expense of the environment is not sustainable [2]. In the pursuit of economic efficiency and increasing output, the use of fertilizers and pesticides has become more intensive [3]. The production mode of high input, high output, and high waste in agricultural systems has destroyed the rural ecological environment, and the country is faced with a growing number of ecological problems [4]. Chinese authorities have issued a series of policy documents, such as “The Opinions on Promoting Agriculture Green Development by Innovating Institutions and Mechanisms” and “The Opinions on Vigorously Implementing the Rural Revitalization Strategy and Accelerating Agricultural Transformation and Upgrading.” These policies are intended to build a high-quality modern agricultural production system and promote the sustainable development of the agricultural economy through scientific and effective agricultural green production methods [5]. The Chinese economy is undergoing a new development mode that emphasizes an environmentally friendly green economy and high living standards [6].

Green agricultural production is the application of sustainable development principles in agriculture, respecting ecological, economic, and social constraints while ensuring food production [7]. Saving energy and reducing pollution are the main goals of agricultural green production [8]. By reducing the use of inputs and improving the utilization of agricultural resources, agricultural non-point source pollution can be controlled, and sustainable agricultural development can be established [9]. The main way to promote agricultural green production is to encourage farmers to adopt high-quality, eco-friendly agricultural green production technology [10].

Farmers are the basic units of agricultural green production, and their behavior directly affects the environment [11]. The research foci of most existing studies have been either the measurement of farmers’ adoption of agricultural green production [12] or factors influencing farmers’ adoption of agricultural green production [13,14,15,16]. Other researchers have considered the impact of farmers’ agricultural green production on the whole social economy and environment [17,18]. However, there is currently no general theoretical framework for the study of farmers’ adoption of agricultural green production. Furthermore, China’s agricultural green production is still in its infancy and needs to be widely promoted [19]. Along with its rapid economic growth, China has faced excessive consumption of resources and environmental pollution that have undermined the overall happiness of mankind [20]. Therefore, although the material achievements (income and environmental protection, etc.) of adopting agricultural green production have been investigated, there are still many areas that need to be explored. One of these areas is the influence of adopting agricultural green production on psychological factors (happiness). Happiness is a topic of interest to wider society [21]. A growing number of countries incorporate happiness into their policymaking and in the measurement of economic benefits and social happiness [22]. When an actor acts, happiness may follow, because their actions have results, one of which is happiness [23]. Hence, economists have been paying more attention to the study of happiness, attempting to explicate the relationship between economic behaviors and happiness [24].

Under the new normal of a harsh “post-COVID” economy, people are facing more challenges and uncertainties [25]. The factors that influence human happiness have become more complex [26,27]. China has about 300 to 400 million farmers, who are the main force supporting agricultural development [28]. As happiness has become an important indicator of the quality of life [29], the happiness of these rural dwellers is related not only to the steady progress of agricultural green production but also determines social harmony and stability to some extent. Researchers and policymakers are eager to know how environmentally friendly behaviors influence the happiness of farmers [30]. Therefore, this paper will try to answer the following three questions: (1) Are farmers happy if they adopt agricultural green production? (2) Which mechanisms adopted by the farmers for agricultural green production affect their happiness? (3) In order to improve the happiness of farmers, what policy proposals should be put forward? The existing literature has not solved the abovementioned problems completely, and empirical evidence is urgently needed.

## 2. Literature Review

### 2.1. Definition of Farmers’ Happiness

Happiness is defined as a state of regularly experiencing positive attitudes, comfort, and satisfaction with life as well as the pursuit of higher life values [31]. The natural human right to which every human being aspires is happiness [32]. It is a felt experience of the overall comprehensive cognition of individuals or families and is an important index to measure people’s quality of life [33]. Happiness is also informed by the way that one considers different aspects of one’s life and can be defined as a subjective expression of personal welfare [34]. Individuals are the ultimate “judges” assessing their life experiences [35]. It includes assessments of life satisfaction, feelings of excitement, positive emotions, and a lack of depression and anxiety [36]. In this study, happiness refers to the various happy feelings experienced by farmers after adopting agricultural green production. This is a state of mental pleasure that arises from achieving what one considers to be good or successful. This is a farmer’s subjective and spiritual feeling, based on memory, which requires subjective judgment [37].

### 2.2. The Influencing Factors of Happiness

Studies have found a U-shaped relationship between age and happiness, with the young and old reporting greater happiness than the middle-aged [38]. Highly educated people feel happier than the less educated, and given the same external conditions, women tend to be happier than men [39]. Additionally, more and more evidence has pointed to the link between high levels of happiness and health outcomes [40]. There is a two-way relationship between happiness and health. Increased happiness leads to an improvement in health, which further improves happiness [41]. Religious belief brings certainty and happiness into one’s life [42]. In a Hungarian study, Lelkes [43] found that religious participation had a positive impact on individual happiness. From a family perspective, entrepreneurial decision-making affects household happiness significantly [44]. The family’s place of residence also significantly affects the happiness of family members. Living in a big city is conducive to access to jobs, employment, and amenities; people who live in urban areas are happier than those who live in rural areas [45]. Happiness is correlated with mean income in one’s neighborhood, and the often lower environmental quality characterizing poorer neighborhoods could contribute to lower well-being in those areas [46]. To sum up, from the perspective of personal and family endowments, happy people are generally described as young, well-educated, female, physically fit, religious, entrepreneurial, and with families living in urban and affluent areas [47,48].

External factors are gaining increasing and explicit focus in the study of happiness. Macroeconomic factors such as inflation, unemployment, and GDP are crucial factors affecting residents’ happiness. Frey et al. [49] found inflation to be the main factor in declining real income and purchasing power, thus negatively affecting happiness. Zhao and Sun [50] found that people have high expectations of happiness in provinces that have high gross domestic product (GDP). When a country’s overall unemployment rate rises, residents’ happiness declines with it [51]. Many studies have looked at the social factors that affect people’s happiness, such as social capital and social trust [52]. Social networks have a significant positive effect on happiness, and individuals who use social networks are on average more satisfied with their lives than those who do not [53]. Government policy is also a prominent factor [54]. Total government expenditure can promote happiness if the expenditure is on health care, public facilities, and other social-security-system-related factors [55]. Commuting time and transport policy affect happiness, and a reduction in commuting time can increase happiness as long commutes have a negative effect on happiness [56].

The existing literature presents a lot of research on the meaning and influencing factors of happiness, which has laid a solid foundation for follow-up research. After broadening the horizons, it can be found that local and foreign studies on factors affecting happiness have focused on cities, whereas research into the happiness of farmers has received scant research attention. Under the development trend of green agriculture, the effect of adopting green agricultural production on farmers’ happiness needs to be further tested. Therefore, based on the survey data of 1138 kiwifruit growers in Shanxi Province, it is undoubtedly meaningful to study the impact of farmers’ adoption of agricultural green production on their happiness and its mechanism. On the one hand, this study theoretically provides a new idea for the study of farmers’ behavior and happiness and makes up for possible deficiencies. On the other hand, it also provides valuable research results for the government to make policies to promote farmers’ happiness in practice.

On the basis of our literature review, the following contents of this paper are arranged as follows. The first section is the introduction while the second section presents a review of the related literature and defines the meaning of core variables. The third section presents the research framework of the relationship between adopting agricultural green production and farmers’ happiness, after which the fourth section presents the data and data analysis methods. In the fifth section, the results are presented and discussed while section six resolves the endogeneity problem and conducts a robustness test. In the seventh section, conclusions are drawn, contributions to knowledge are identified, policy suggestions are made, and limitations are stated to identify areas needing further research.

## 3. Theoretical Analysis and Hypothesis

In economics, happiness is often portrayed as utility. In psychology, it is referred to as subjective welfare [57]. Both reflect one’s subjective evaluation of one’s own life as a whole [58]. A country should not just be concerned with economic progress but also the emotions of its residents. This is a significant topic worthy of attention and investigation. In this paper, our focus is on the link between the adoption of agricultural green production and farmers’ happiness(Figure 1).

In a survey of farmers from northern Ghana, Martey et al. [59] found that green agricultural practices significantly increased land productivity and agricultural income by 55% and 30%, respectively. Similarly, Donkor et al. [60] found that green agricultural processes increased cassava yields and increased farmers’ net revenues in Nigeria. Green agriculture is the future of agricultural development and has obvious economic benefits. It enhances the competitiveness of agricultural products, promotes agricultural efficiency, and increases farmers’ incomes. Hence, adopting agricultural green production can lead to an increase in absolute income. Using data from the U.S. and 11 other countries to compare happiness across countries, American economist Robert Easterlin proposed the well-known “paradox of happiness”: in the early stages of economic growth, increasing income translates to increased happiness. After a certain income level, happiness decreases with increasing income, that is, income growth has a reverse U-shaped relationship with happiness [61]. Nevertheless, subsequent happiness research has shown that, on average, the happiness of the rich is still greater than that of the poor within a country [62]. In neoclassical economics, happiness is closely related to utility. Higher income represents access to more choices to better meet personal preferences and needs, which means higher income can bring greater happiness. In developing countries, there is a positive correlation between income growth and happiness for low-income people [63]. Chinese farmers are incurring debts on account of the rising cost of basic needs (such as human association, education, and medical treatment). A higher income helps farmers repay their debts, bringing a sense of relief and happiness. In other words, the marginal contribution of the increase in absolute income to happiness is significant for Chinese farmers. Engaging in agricultural green production is one way to achieve higher incomes, and consequently, happier outcomes.

The first hypothesis can be stated as follows:

**H1:** *Adopting agricultural green production promotes farmers’ happiness by increasing absolute income*.

The emergence of green consumption can drive market reform, and those engaged in green production can corner this market. Farmers who adopt agriculture green production can earn higher incomes than ordinary farmers. Hence, adopting agriculture green production can increase the relative income. Alpizar et al. [64] adopted experimental methods to study the influence of people’s comparative behavior on happiness, while the comparison of income is an economic factor that may influence happiness [65]. The existence of the relative deprivation effect will decrease people’s evaluation of their own happiness with the increase in others’ income [66,67]. Paul and Guilbert [68] suggested that people derive utility by comparing themselves with a reference group. In a Bangladeshi study, respondents reported higher satisfaction levels when their income increases over time, but people whose incomes were lower than that of their neighbors’ reported less satisfaction with life [69]. Thus, one’s happiness depends on one’s own income and that of a reference group [70,71]. In view of these, the higher the relative income, the happier the individual [72]. Engaging in agricultural green production increases the relative income of farmers, which leads to their happiness.

Using this argument, we arrive at our second hypothesis, which is as follows:

**H2:** 
*Adopting agricultural green production promotes farmers’ happiness by increasing their relative income.*


Agricultural green production technologies such as organic fertilizer substitution technology and green pest control technology reduce soil and air pollution caused by the excessive use of chemical products [73]. Agricultural green production technologies such as water-saving irrigation are another win–win management strategy in that they reduce greenhouse gas emissions and improve water use efficiency [74,75]. Farmers’ adoption of agricultural green production helps to mitigate environmental pollution and conserve resources, with significant ecological effects. As more farmers go green, a healthy ecological environment as the fairest public good is a beneficial outcome. Pollution affects people’s work and life as well as their health, negatively affecting many measures of happiness [76,77]. Does this imply that reducing pollution increases happiness? The answer is a firm yes [78]. Pollution abatement is expected to improve the quality of the local environment and thus enhance people’s happiness [79]. As engaging in agricultural green production improves the habitable environment by alleviating environmental pollution, more happiness follows [80].

Based on these reasons, we state our third hypothesis as follows:

**H3:** 
*Adopting agricultural green production promotes farmers’ happiness by mitigating agricultural pollution.*


There are five levels in Maslow’s hierarchy of needs. These are from the lowest to the highest: physiological needs, security needs, social needs, self-esteem needs, and self-actualization needs. After the most basic needs have been satisfied, people begin to focus on higher needs such as spiritual needs [81]. Hence, some scholars have suggested that farmers switch to agricultural green production not only to increase their income but to benefit from the advantages that come with the change in social status [82]. Along with the increasing specification of villagers’ autonomy, village open elections have become an important source of power, and farmers engaged in green agricultural production may come across as stronger. They may also have more opportunities to become party members or belong to the leading leaders, thus improving their status in the village. Many empirical studies have investigated the link between social status and happiness. Van et al. [83] found that higher-ranking castes are more satisfied than lower- and middle-ranking castes. There was an indirect and positive relationship between social status and happiness through job, family, and income satisfaction. Social status and changes in social status have a significant impact on happiness, and a positive perception of social status has a positive impact on individual happiness. Engaging in agricultural green production can promote farmers’ happiness by enhancing their social status in the village.

Therefore, our fourth hypothesis is stated as follows:

**H4:** 
*Adopting agricultural green production promotes farmers’ happiness by elevating their social status.*


The above analysis shows that farmers’ engagement in agricultural green production can increase their absolute and relative income, mitigate agricultural pollution, and elevate their social status to improve their happiness. From what we discussed above, we arrive at a fifth hypothesis, which is:

**H5:** 
*Adopting agricultural green production has a significant positive effect on the happiness of farmers.*


## 4. Materials and Methods

### 4.1. Source of Data

Compared with food crops, fruits have higher economic benefits, and the amount of fertilizer is generally higher. China produces and consumes large quantities of kiwi fruits. This study used data from a field survey of kiwi growers in Shanxi Province in 2022. The reason for choosing Shanxi Province as the study area is that the development of the kiwifruit planting industry in this area is representative. The scale of the kiwifruit industry in Shanxi accounts for about 40% of the whole country, ranking first in China. It is recognized by experts as the best origin and concentrated producing area of kiwifruit. To ensure the representativeness of research samples, this paper selected Zhouzhi County, Mei County, Wugong County, and Yangling District (Figure 2). The sample for this survey was selected by stratified sampling and random sampling. Four villages were then randomly selected from each and 20–25 farmers were randomly chosen from each village. The survey was conducted through face-to-face interviews. Overall, 1200 questionnaires were distributed. After eliminating the questionnaires with missing and incomplete data or those with logical errors, 1138 valid and complete questionnaires remained, an effective response rate of 94.83%.

### 4.2. Descriptive Analysis of Farmers’ Happiness and Adoption of Agricultural Green Production

A core research variable of this research is farmers’ happiness. The distribution is shown in Figure 3. Of the 1138 respondents, 151 farmers were “very happy “ and 443 farmers were “happy“, accounting for 13.27% and 38.93% of the total sample, respectively; 468 farmers with “general happy“, accounting for 41.12%. There were 57 “uhappy“ farmers and 19 “very unhappy“ farmers, accounting for 5.01% and 1.67% of the total sample, respectively (Figure 3).

Another core research variable of this study is the adoption of agricultural green pro-duction by farmers. The distribution is shown in Figure 4. A total of 168 farmers (14.76%) adopted water-saving irrigation technology while 566 farmers (49.74%) utilized green pest control technology. A total of 778 households (68.37%) adopted organic fertilizer substitution technology. In general, the relative adoption rates of the three technologies are: organic fertilizer substitution technology > green pest control technology > water-saving irrigation technology.

The association between farmers’ adoption of agricultural green production and their happiness is represented in Figure 5. A total of 77.94% of the sampled farmers implemented at least one of the three agricultural green production technologies. In this group of farmers, 0.34% of households regarded themselves to be very unhappy, 1.13% of households considered themselves to be unhappy, 40.25% felt generally happy, 44.64% felt happy, and 13.64% felt very happy. Amongst the respondents, 22.06% of farmers had not adopted any of the three agricultural green production technologies. In this group of farmers, 6.73% of farmers considered themselves very unhappy, 18.37% of farmers felt unhappy, 44.22% of farmers felt generally happy, 18.73% of farmers felt happy, and 11.95% of farmers felt very happy. The proportion of happy farmers in these two groups demonstrates that those who implemented environmentally friendly agricultural practices were happier than those who did not.

### 4.3. Variables

Data were collected using questionnaires that assessed the happiness of farmers, farmers’ adoption of AGP, mediation variables, and control variables. As the COVID-19 pandemic has been going on in China for more than three years [84], the current study may have been limited in terms of the depth and breadth of data. Panel data were not used for the study. However, the research enriched questionnaire questions in the investigation and measurement methods of variables as much as possible to reduce inaccurate results caused by insufficient sample size. At the same time, we did not set an upper line on the number of farmers surveyed but continued to interview farmers when possible until we found it difficult to obtain new ideas [85]. This also indicates that the data of 1138 farmers have reached data saturation to some extent, which is sufficient to support the needs of this study.

#### 4.3.1. Measuring Farmers’ Happiness

Happiness is one’s overall assessment of the quality of life according to one’s own criteria [86]. Measuring this broad and all-encompassing spectrum of human happiness is difficult because it cannot be measured independently [87]. Since people can rate their own happiness consistently and without confusion, many fields of research use relevant scales to identify happiness variables by asking respondents about their subjective feelings [88]. In the World Values Survey (WVS) conducted by Ronald Inglehart, a political science professor at the University of Michigan [89], the questionnaire measures happiness by asking the question “In general, do you feel happy?”. Similarly, the General Social Survey and Gallup World Poll Surveys also use questionnaires to quantify happiness: respondents rate themselves on a scale of life satisfaction by selecting between options “very happy,” “happy,” “not very happy,” and “not at all happy” [90,91]. The quality of life instrument is a horizontal VAS measuring happiness. It has demonstrated favorable psychometric properties, using a range from 0 (“completely unhappy”) to 10 (“completely happy”) [92]. Although the methods above are relatively simple, existing studies have shown that the indices measured by these methods have comparably high validity and reliability. Thus, they can accurately express the true feelings of individuals [93].

Several studies have pointed out that happiness consists of both an emotional and cognitive component [94,95]. Emotional components include pleasant emotions and negative emotions. It can be measured by the Positive and Negative Affect Scale (PANAS) [96]. The higher the level of happy emotion and the lower the level of negative emotion, the higher the happiness [97]. The cognitive part represents the individual’s satisfaction with their current state of life after a comprehensive assessment. It can be measured by the Life Satisfaction Scale [98]. The higher the satisfaction, the higher the happiness [99].

Drawing on the above methods, the emotional and cognitive components could help us to achieve our goal of getting closer to the meaning of happiness [100]. The questionnaire in this study included the questions:Emotional component:
“Do you feel joy for most of the year?”“Do you feel sad for most of the year?”Cognitive component:
“Are you satisfied with your life in recent years?”

Using a 5-point Likert scale, we assigned 1 = little; 2 = very little; 3 = indifferent; 4 = strong; and 5 = very strong [101]. Emotional and cognitive components are used to calculate the weighted average, which is the happiness of farmers.

#### 4.3.2. Measuring Farmers’ Adoption of Agricultural Green Production

This study defines engagement in agricultural green production by farmers as the adoption of one or more technologies to reduce agricultural pollution in the planting process. The behavior of selected kiwifruit growers throughout the kiwi production cycle was investigated, with emphasis on the technologies used in irrigation, pesticide, and fertilizer application. The three agricultural green production technologies selected are as follows: (1) water-saving irrigation technology is a kind of ecological environment protection technology to improve the utilization rate of water resources and alleviate the crisis of water resources; (2) organic fertilizer substitution technology is a kind of technology that can improve the quality of agricultural products and land pollution by using organic fertilizer; (3) green pest and control technology is a pesticide application technology that integrates physical control, biological control, and ecological control [102].

This study evaluated farmers’ adoption of agricultural green production in two ways. The first was to determine whether farmers had engaged in agricultural green production. The second was a comprehensive evaluation of farmers’ adoption of agricultural green production. In the questionnaire, farmers were asked three questions: “Did you adopt water-saving irrigation technology (use water-saving irrigation equipment)?”, “Did you adopt organic fertilizer substitution technology (animal waste including animal manure, animal processing waste, and plant residues including cake fertilizer, crop straw, fallen leaves, dead branches, and grass charcoal as fertilizer)?”, and “Did you adopt green pest control technology (the following methods were used to kill the insects: light, yellow plate, sweet and sour liquid, sex bait, artificial killing, and belt trapping)?”. Each action was set as a binary variable; the variable was assigned a value of 1 if the action was performed and 0 if it was not. Farmers were considered to have engaged in agricultural green production if they had engaged in any of the three agricultural green production technologies. By summing up the values of the three variables, the comprehensive value of each farmer’s adoption of agricultural green production was obtained. If no agricultural green production technology was adopted, the value was 0; if one agricultural green production technology was adopted, the value was 1; if two agricultural green production technologies were adopted, the value was 2; if three agricultural green production technologies were adopted, the value was 3.

#### 4.3.3. Measuring Mediation Variables

According to the research hypotheses and mechanism analysis, the mediation variables are increasing absolute income, increasing relative income, mitigating agricultural pollution, and elevating the social status. Using a 5-point Likert scale, extremely effective was assigned a score of 5; very effective, a score of 4; effective, a score of 3; less effective, a score of 2; and no change, a score of 1.

#### 4.3.4. Control Variables

Based on the existing happiness research literature, the following control variables were added to the empirical analyses to minimize the problem of missing variables: age, education, laborers, marital status, health condition, cooperative member, and government worker [103,104,105]. Table 1 defines and explains the variables in the model selected for the study and illustrates the descriptive statistics.

### 4.4. Model Specification

#### 4.4.1. Ordered Probit Model

The happiness value is a typical discrete variable that is ordered. Therefore, the ordered probit model was utilized to assess the effect of adopting agricultural green production on their happiness:(1)$$\begin{array}{c} Y_{i}{}^{*}=\alpha_{1}X_{i}+\alpha_{2}C_{i}+\varepsilon_{i}\;\; \end{array}$$

In Equation (1), $Y_{i}{}^{*}$ represents the happiness of farmers, $X_{i}\;$ indicates farmers’ adoption of green agricultural production, $C_{i}\;$ is the control variables that affect the happiness of farmers$,\;\alpha_{1}$ and$\;\alpha_{2}$ are the parameter to be estimated, and$\;\varepsilon_{i}$ is a random disturbance term.

#### 4.4.2. Mediation Effect Model

The mediation effect model was used to determine how engagement in agricultural green production influences farmers’ happiness by increasing their absolute and relative income, mitigating agricultural pollution, and elevating their social status [106]. At each step of the examination, the OLS model was utilized. Testing the correlation between the adoption of agricultural green production and the happiness of farmers was the initial step. If the estimation result was significant, additional verification was conducted based on the effect of mediation. The second phase consisted of testing the link between the adoption of agricultural green production and the mediating variables. In the third step, the regression model of farmers’ adoption of agricultural green production, farmers’ happiness, and intermediate variables was constructed to verify the intermediate variables. If both the estimation results of Equations (3) and (4) are significant, then the indirect effect is considered to be significant. If one of the estimation findings is not statistically significant, the bootstrap test is utilized. Rejection of the null hypothesis shows that the indirect effect is significant.

The equations are as follows:(2)$$\begin{array}{c} \;\;Y_{i}=\alpha_{1}X_{i}+\alpha_{2}C_{i}+\varepsilon_{i}\;\; \end{array}$$
(3)$$M_{i}=\alpha_{0}+\alpha_{3}X_{i}+\alpha_{4}C_{i}+\varepsilon_{i}\;$$
(4)$$\begin{array}{c} Y_{i}=\alpha_{5}X_{i}+\alpha_{6}M_{i}+\alpha_{7}C_{i}+\varepsilon_{i}\;\; \end{array}$$

In Equations (3)–(5), $M_{i}\;$ is the mediating variable, including increasing their absolute and relative income, mitigating agricultural pollution, and elevating social status. $\alpha_{0},\;\alpha_{1},\;\alpha_{2},\;\alpha_{3},\alpha_{4},\alpha_{5},\alpha_{6},\alpha_{7}\;$ is the parameter to be estimated. The meaning of the remaining variables and symbols is the same as that of Equation (1).

## 5. Results and Discussion

### 5.1. The Direct Effect of Adopting Agricultural Green Production on Farmers’ Happiness

In this section, the Order Probit Model was used; Table 2 shows the estimation results using Stata 14.0. Model (1) demonstrates that adopting agricultural green production has a considerable beneficial effect on farmers’ happiness and that the happiness of farmers who engage in agricultural green production is 28.6% more than that of farmers who do not. According to the empirical findings of Model (2), the comprehensiveness with which agricultural green production technologies are embraced has a favorable effect on farmers’ happiness. The marginal effect was 14.6%, which was statistically significant at the 1% significance level. Thus, it can be seen that the more kinds of technologies that are adopted, the better the happiness of the farmers. Models (3), (4), and (5) show that the adoption of water-saving irrigation technology, green pest control technology, and organic fertilizer substitution technology by farmers can increase their happiness, and all three pass the statistical significance test at the 1% level. The marginal effects of the three distinct agricultural green production technologies on farmers’ happiness varied. Adopting water-saving irrigation technology has the greatest marginal effect of the three, at 30.3%. The marginal effect of organic fertilizer substitution technology is 27.4%, whereas the marginal effect of green pest control technology is 19.8%. China’s water-saving irrigation systems are quasi-public commodities and are mostly sponsored by the government, resulting in the greatest marginal effect of water-saving irrigation technology. Fewer inputs and greater outputs make it simpler for farmers to achieve contentment. Green pest control technology has the lowest marginal effect, most likely because it is difficult to master and requires extensive application to be effective.

In terms of control variables, there is a “U” shaped relationship between farmers’ age and happiness. The possible reason is that farmers have less life pressure when they are young. Later, due to the increasing pressure of work, life, and finances, their happiness decreases with an increase in age. Gradually, the life of farmers begins to be comfortable and stable. When they reach a certain age, their happiness gradually increases with an increase in age. The happiness of married farmers is significantly higher than that of single farmers, and marriage enriches the family life of farmers, thus promoting the improvement of happiness. The better the health status, the higher the happiness of farmers. At the same time, the number of household members in the labor force also has a significant positive impact on the happiness of farmers. More family labor force reduces everyone’s working hours and also improves the happiness of farmers from a certain point of view. Working in government and participating in agricultural cooperatives have significant positive effects on farmers’ happiness. Surprisingly, higher levels of education did not improve farmers’ happiness as much as expected. The possible reason is that higher education may bring more ideas, but they cannot be implemented due to the limited rural resources.

### 5.2. Heterogeneity Analysis

If the use of agricultural green production technology is extended to the whole farm, will it have a greater impact on farmers’ happiness? In order to solve this problem, this study divided the types of surveyed farmers into a small farmers group (group 1) and new agricultural operators group (Group 2) for in-depth analysis. New agricultural operators can effectively achieve specialized and large-scale production and operation [107,108]. According to the characteristics of production and operation, new business entities can be divided into four types: large professional households, family farms, farmers’ professional cooperatives, and agricultural enterprises [109,110].

Large professional households are mainly specialized in a specific agricultural industry, and their production scale is larger than that of ordinary farmers, reaching the scale standard determined by the agricultural department.

Family farms refer to those with family members as the main labor force, with a relatively high degree of intensification and specialization, and an area of more than 50 mu. They master agricultural production methods and technologies and have certain financial strength and production conditions.Farmers’ professional cooperatives refer to the mutual-aid economic organizations in which farmers who produce the same kind of agricultural products voluntarily cooperate through land, labor force, capital, and technology.Agricultural enterprises refer to the adoption of modern enterprise management methods and professional division of labor and cooperation. It is engaged in the whole industrial chain of planting and processing, warehousing, logistics, transportation, sales, and even scientific research of agricultural products. Economic organizations operate independently and assume sole responsibility for their profits or losses.

The specific regression results are shown in Table 3. The empirical analysis results of Models (1) and (2) show that the adoption of agricultural green production can significantly improve the happiness of farmers in both the small farmers group and the new agricultural operators group. However, the happiness effect of agricultural green production adoption was higher in the group of new agricultural operators compared to in the group of small farmers. The empirical analysis results of Models (3) and (4) show that the more types of agricultural green production technologies that are adopted, the more effectively the happiness of new agricultural operators can be improved. For a single technology, from the empirical analysis results of models (5)–(10), it can be seen that the happiness of new agricultural operators adopting water-saving irrigation technology, green pest control technology, and organic fertilizer substitution technology is higher than that of ordinary small farmers.

### 5.3. The Mediation Effect of Adopting Agricultural Green Production on Farmers’ Happiness

The mediation effect model was used to assess whether farmers’ adoption of agricultural green production affects their happiness by increasing their absolute and relative income, mitigating agricultural pollution, and elevating their social status. In Table 4, Model (1) shows that with respect to the overall impact, adopting agricultural green production has a positive impact on farmers’ happiness. Models (2)–(5) show that adopting agricultural green production increases the absolute and relative income, mitigates agricultural pollution, and elevates the social status. Models (6)–(9) show that increasing the absolute income and relative income, mitigating agricultural pollution, and elevating the social status have a significant positive impact on the happiness of farmers. Model (10) shows that including all variables still produces a significant result. By creating models comparing independent variables to dependent variables, independent variables to mediation variables, and independent and mediation variables to dependent variables, a partial mediation effect was identified.

The influence of agricultural green production on the happiness of farmers is positively significant at a 1% significance level, with a coefficient of 0.455. After introducing the mediation variables, the influence of agricultural green production on the happiness of farmers is still positively significant, with a coefficient of 0.342, 0.213, 0.199, and 0.220, respectively. Therefore, the four variables increasing absolute income, increasing relative income, mitigating agricultural pollution, and elevating social status have a partial mediating effect on the relationship between the adoption of agricultural green production and farmers’ happiness, with the mediation effects being 11.3%, 24.2%, 25.6%, and 23.5%.

## 6. Robustness Testing and Analysis of the Endogeneity Problem

### 6.1. Robustness Testing

#### 6.1.1. Replace the Explained Variable Assignment

In this study, the measurement of variables is the judgment made by the subjective evaluation of farmers. Due to the difference in farmers’ own cognitive levels, there may be some errors in data measurement. For example, farmers may overstate or understate their happiness, but it is impossible to judge who is in what kind of situation during the survey [111]. Therefore, this study uses the method of replacing the explanatory variable assignment to test the robustness [112]. The following methods are used to re-evaluate the happiness of farmers and estimate it by using the binary probit model.

Understate happiness: merge “very good”, “good” and “average” into “good” and assign a value of 1, and merge “bad” and “very bad” into “bad” and assign a value of 0.Overstate happiness: merge “very good” and “good” into “good” and assign a value of 1, and merge “bad”, “very bad” and “average” into “bad” and assign a value of 0.

According to the results in Table 5, regardless of the situation of “understate happiness” or “overstate happiness”, the adoption of agricultural green production still has a significant impact on farmers’ happiness. It shows that the estimation results of the model are robust.

#### 6.1.2. Confidence Interval Analysis

When we use a sample to estimate the population, we use a range of methods to make sure that the sample is unbiased so that the sample is as representative as possible. However, this sample is still not 100% representative of the population, and there are always errors. The confidence interval represents the interval of the range of sample estimated population mean, and the population information is estimated by sample information. The commonly used confidence level is 95%, which guarantees that 95% of the sample mean will fall within two standard errors. The higher the confidence level, the wider the interval, and the greater the probability that the confidence interval contains the population mean. Lower confidence interval = population mean-|Z| standard errors. Upper confidence interval = population mean + |Z| standard errors. The data in this study accord with the principle of statistics. In this paper, the correlation between the key variables in the model is tested. The test is judged by whether the confidence interval at the 95% confidence level contains “0” [113].

If the confidence interval contains a “0” value, that is, the lower limit is negative and the upper limit is positive, the effect is not significant.If the confidence interval does not contain a value of “0” and both the lower and upper limits are positive, the effect is significant and positive.If the confidence interval does not contain the value “0” and the lower limit and upper limit are both negative, the influence is significant and negative.

According to the results in Table 6, at the 95% confidence level, the adoption of green production still has a significant impact on farmers’ happiness. It shows that the estimation results are reliable.

#### 6.1.3. Reanalysis of Mediating Effects in Partial Samples

Considering the physical condition of the elderly and their ability to accept technology, this paper excluded the samples of the elderly over 60 years old and re-analyzed the mediating effect. The results are shown in Table 7. The effect of adopting agricultural green production on farmers’ happiness was positive and significant at a 1% significance level, with a coefficient of 0.557. After introducing the mediating variable, the effect of adopting green agricultural production on farmers’ happiness is still positive and significant, and the influence coefficients are 0.386, 0.337, 0.331, and 0.326, respectively. Therefore, the four variables increasing absolute income, increasing relative income, mitigating agricultural pollution, and elevating social status have a partial mediating effect on the relationship between the adoption of agricultural green production and farmers’ happiness, with the mediation effects being 17.1%, 22.0%, 22.6%, and 23.1%, respectively. The regression results of the partial samples are basically consistent with those of the whole samples. The above evidence shows that the subjective answers of the respondents are not the major problems faced by this paper in the case of survey data from 1138 farmers.

### 6.2. Analysis of the Endogeneity Problem

The results so far have shown that the happiness of farmers is increased by their engagement in agricultural green production. However, there is a causality question: do farmers who engage in agricultural green production become happier, or do farmers who are happy engage in agricultural green production? Since farmers’ happiness is a comprehensive result of a variety of subjective and objective factors and some factors are difficult to measure, it is inevitable that some important variables are omitted. To minimize the adverse effects of endogeneity, the propensity score matching (PSM) method was used to mitigate sample selection and address self-selection and endogeneity problems [114]. To achieve this, the respondent farmers were divided into an experimental group and a control group. Farmers who engaged in agricultural green production (experimental group) were matched with those who did not (control group).

The average treatment effect ATT can be written as:(5)$$ATT=E\left(Y_{1}/L_{m}=1\right)-E\left(Y_{0}/L_{m}=1\right)=E\left(Y_{1}-Y_{0}/L_{m}=1\right)$$

In this formulation, $Y_{1}$ is the happiness of farmers who have adopted green agricultural production, and $Y_{0}$ is the happiness of farmers who have not adopted green agricultural production. *E*($Y_{1}/L_{m}=1$) can be predicted, while *E*($Y_{0}/L_{m}=1$) cannot be predicted directly. This is the counterfactual result. The calculation of the weight function depends on the matching method. In this study, the methods used to calculate the net effect of treatment were nearest neighbor matching and kernel matching.

#### 6.2.1. Balance Test

The scores of the respondents were matched using the nearest neighbor matching and kernel matching methods. The results are listed in Table 8. The value of Pseudo declined substantially from 0.486 to 0.112–0.134, as did the values generated by the likelihood ratio statistic: from 549.68 to 196.91–266.68. Both changes indicate that the PSM method reduced the differences between the experimental and control groups in the explanatory variables and brought down the deviation of sample selection significantly, with an acceptable matching quality [115].

#### 6.2.2. Effects of the Treatments

As Table 9 shows, when there is no matching, the average treatment effect of the total sample is 0.510, and the t-value is 8.73, which is statistically significant at the 1% significance level. Four matchings were used to estimate the ATT and check the accuracy. With kernel matching (KN) (0.06), the marginal effect of not engaging in agricultural green production is 3.138. The marginal effect of engaging in agricultural green production is 3.625. It can be seen that engaging in agricultural green production can significantly promote the happiness of farmers. The other three matching methods can be proved similarly. Therefore, the above empirical analysis results have good robustness.

## 7. Conclusions and Policy Implications

### 7.1. Findings

China is a country with a substantial agricultural industry, which is vital to the national economy and socioeconomic activities. Using microscopic survey data from 1138 kiwi farmers in the major kiwi-producing regions of Shanxi Province, China, this study examines how farmers’ participation in agricultural green production affects their happiness. The three primary conclusions are as follows: first, engaging in agriculture green production increases the happiness of kiwi growers by 28.6%; through group study, it can be found that adopting agricultural green production can significantly improve the happiness of both small farmers and new agricultural operators. At the same time, the happiness effect of agricultural green production adoption was higher in the group of new agricultural operators than in the group of small farmers. Second, three types of agricultural green production technologies raise the happiness of farmers by a large amount. The marginal effect of water-saving irrigation technology is the greatest, followed by organic fertilizer substitution technology and green pest control technology. Those who adopt a greater variety of technologies are happier than those who adopt a lesser variety. Third, increasing absolute income and relative income, mitigating agricultural pollution, and elevating social status partially mediate the association between the adoption of agricultural green production and farmers’ happiness, with respective effects of 11.3%, 24.2%, 25.6%, and 23.5%. In this paper, the replacement of the explained variable assignment, confidence interval analysis, and the reanalysis of mediating effects in partial samples were conducted for the robustness test. In addition, propensity score matching (PSM) was used to solve endogeneity problems (Figure 6).

### 7.2. Policy Implications

These findings confirm that the adoption of agricultural green production significantly improves farmers’ happiness. According to our findings, the following recommendations are made.

To increase farmers’ happiness, agricultural green production should be promoted. However, for green agricultural development to result in sustainable development, an effective model suitable for the basic management systems of rural China and the decentralized management of farmers must be in place. At the macro level, it is necessary to optimize the agricultural industry system and remodel the green production system. At the micro level, measures should be taken to encourage farmers to adopt agricultural green production. An education program should be devised with the goal of making farmers realize that it is advantageous for them to adopt agricultural green production and that the government wants to help them to do so in order for them to have a healthier and more eco-friendly lifestyle in rural areas. The government, leading enterprises, and scientific research institutions can cooperate with farmers to build experimental fields and demonstration parks for the promotion of green agriculture. Channels of communication between farmers and agricultural green production technology experts should be established. This will serve as a platform to disseminate technical knowledge to better engage farmers and equip them with the knowledge needed for agricultural green production. Since different agricultural green production technologies have different effects on farmers’ happiness, the government should strengthen the guidance of farmers to engage in different kinds of agricultural green production technologies either through compulsory means such as legislation or through motivation, by expanding the scope of subsidies and providing non-material subsidies.

The Chinese government should emplace a quality certification system for agricultural green production without delay. This will ensure that the market value of agricultural green products increases and the prices of high-quality green products become more competitive. Effective and efficient links should be provided free of charge between the green produce and the market so that farmers do not have to worry about selling their produce, thus raising the farmers’ absolute and relative income. In addition, participation in agricultural green production could become a prerequisite for the election of village leaders and cooperative leaders as well as access to various subsidies. In summary, once the obstacles to farmers engaging in agricultural green production are removed, not only will farmers enjoy the increased economic and ecological benefits but they will also be better self-actualized and thus feel happier. This will make farmers truly enjoy the economic benefits, ecological benefits, and better self-realization of “happiness fruits” brought by the adoption of agricultural green production.

### 7.3. Possible Contributions to Knowledge

The contributions of this paper to the literature are in three main areas. Above all, behavioral economics and social psychology theories are applied in the study to conduct empirical research on farmers’ economic behavior and their happiness, which expands and enriches the research on farmers’ economic behavior and happiness in a Chinese context. Furthermore, this paper not only studies the overall impact of engaging in agricultural green production on the happiness of farmers but also revealed and tested the mechanism between the adoption of agricultural green production and the happiness of farmers by using the mediating effect method. Last but not least, many studies have used very large data samples, ignoring variables such as policy, regional difference, culture, the natural environment, and the plantation type to control for the inherent differences. However, the data for this study came from 1138 households of farmers from the same region with similar characteristics, such as being subject to the same culture and climate and planting the same crop. As such, the influence of exogenous factors on the happiness of farmers is minimized and the results are more robust. Additionally, the propensity score matching method was used to further verify the causal relationship between adopting agricultural green production and farmers’ happiness, which makes the results more reliable.

### 7.4. Limitations and Areas for Further Research

This paper is only an introductory exploration of the research on the relationship be-tween farmers’ adoption of agricultural green production and happiness. There are still many shortcomings to this research. Although the samples are random and representative to some extent, the data are mainly from Shanxi Province. Farmers who have different life experiences and are from different regions may have different levels of economic development. Thus, there may be significant differences in farmers’ behavior and decision-making. Therefore, the applicability of the research conclusions to other areas needs to be verified. Limited by the data, the discussion of the relationship between farmers’ adoption of agricultural green production and happiness was analyzed at cross-sectional time points, while farmers’ adoption decisions of agricultural green production are dynamic. It is possible that some factors may not have been captured and measured. These are some of the limitations of this study, which are worthy of consideration in future research.

## Figures and Tables

**Figure 1 ijerph-20-02856-f001:**
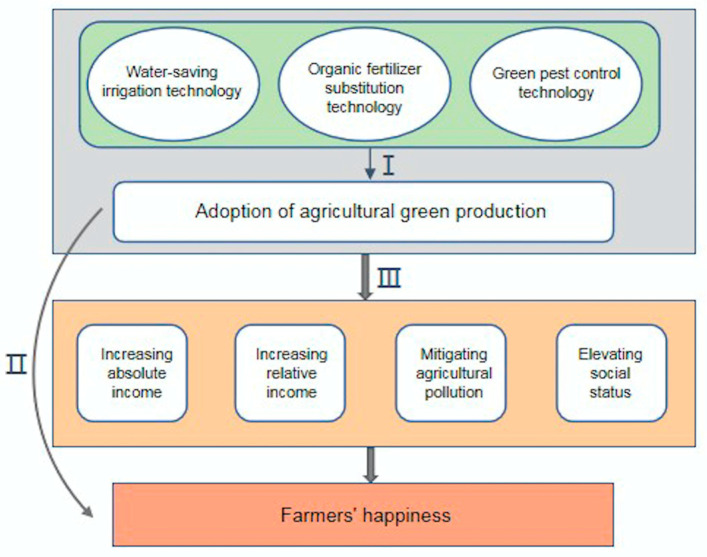
Theoretical model of the study. I represents the three component dimensions of agricultural green production. II represents the direct effect of adopting agricultural green production on farmers’ happiness. III represents the mediation effect of adopting agricultural green production on farmers’ happiness. The light orange color represents the four mediating variables.

**Figure 2 ijerph-20-02856-f002:**
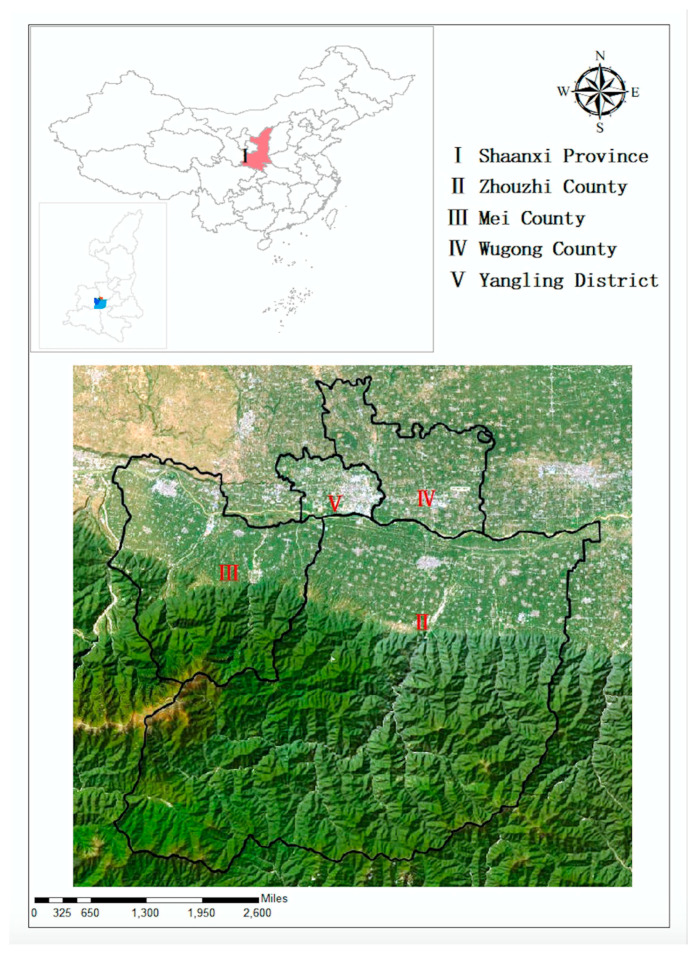
Location of the study sites in Shanxi, China.

**Figure 3 ijerph-20-02856-f003:**
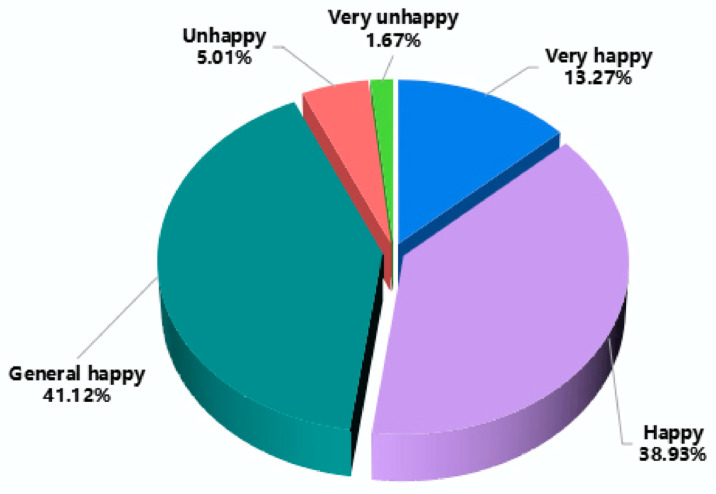
Distribution of farmers’ happiness.

**Figure 4 ijerph-20-02856-f004:**
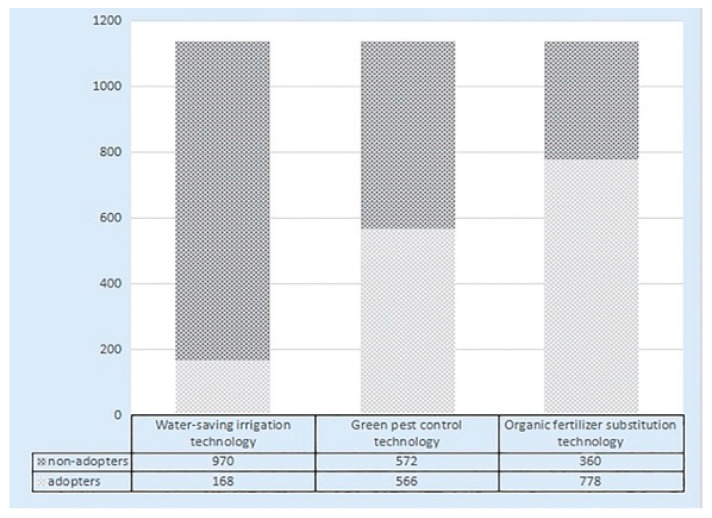
Statistics of farmers’ adoption of agricultural green production.

**Figure 5 ijerph-20-02856-f005:**
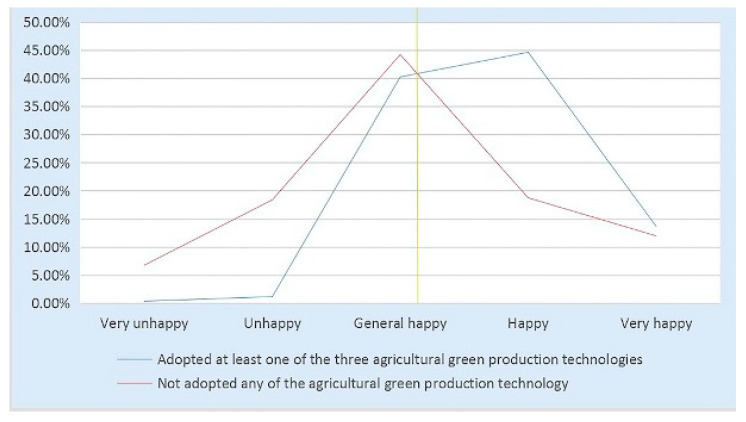
Relationship between adoption of agricultural green production by farmers and their happiness.

**Figure 6 ijerph-20-02856-f006:**
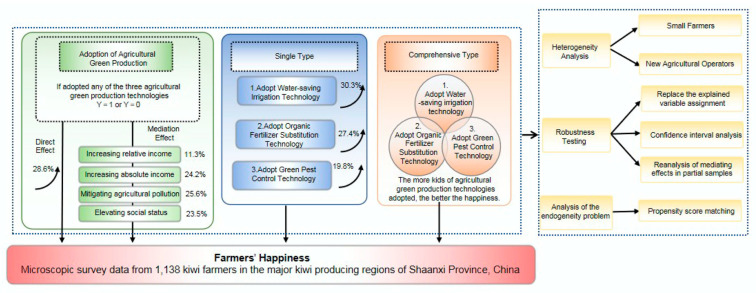
Findings of the manuscript.

**Table 1 ijerph-20-02856-t001:** Variable definition and descriptive statistics.

Variable	Definition	Mean	Standard Deviation
Adoption of Agricultural Green Production	If farmers had engaged in any of the three agricultural green production technologies. Yes = 1, No = 0	0.773	0.419
Comprehensiveness	The number of farmers that had adopted agricultural green production technologies.	1.996	2.341
Happiness	Weighted mean of emotional component and cognitive component.	3.415	—
Emotional components	Do you feel joy for most of the year?	3.847	1.849
	Do you feel sad for most of the year?	2.799	1.653
Cognitive components	Are you satisfied with your life in recent years?	3.600	1.956
Age	Actual age	40.279	36.500
Education	Years of schooling	6.686	2.421
laborers	Number of laborers	2.708	1.091
Marital status	Married = 1, Single = 0	0.713	1.101
Health condition	Very good = 5, Good = 4, General = 3 Bad = 2, Very bad = 1	3.367	1.025
Cooperative member	Yes = 1, No = 0	0.439	0.496
Government worker	Yes = 1, no = 0	0.146	0.353
Increasing absolute income	Extremely effective = 5, Very effective = 4, Effective = 3, Less effective = 2, No change = 1	3.367	1.205
Increasing relative income		3.362	1.171
Mitigating agricultural pollution		2.375	1.314
Elevating social status		3.355	1.025

**Table 2 ijerph-20-02856-t002:** Estimation results of the direct impact of adopting agricultural green production on the happiness of farmers.

Variable	Model (1)	Model (2)	Model (3)	Model (4)	Model (5)
	Happiness				
Adoption of agricultural green production	0.286 *** (0.096)	—	—	—	—
Comprehensiveness	—	0.146 *** (0.081)	—	—	—
Water-saving irrigation technology	—	—	0.303 *** (0.100)	—	—
Green pest control technology	—	—	—	0.198 *** (0.087)	—
Organic fertilizer substitution technology	—	—	—	—	0.274 *** (0.090)
Age	−0.056 *** (0.009)	−0.033 ** (0.010)	−0.051 *** (0.009)	−0.043 *** (0.007)	−0.044 * (0.010)
Age squared	0.032 *** (0.016)	0.100 *** (0.023)	0.031 * (0.010)	0.010 *** (0.009)	0.023 *** (0.007)
Education	0.076 (0.059)	0.066 (0.044)	0.053 (0.042)	0.053 (0.042)	0.057 (0.044)
laborers	0.042 *** (0.016)	0.041 *** (0.014)	0.042 *** (0.016)	0.044 *** (0.017)	0.050 *** (0.023)
Marital status	0.154 ** (0.056)	0.157 *** (0.060)	0.157 *** (0.060)	0.154 *** (0.056)	0.165 *** (0.062)
Health condition	0.195 *** (0.052)	0.185 ** (0.050)	0.195 *** (0.052)	0.120 *** (0.060)	0.186 * (0.051)
Cooperative member	0.321 *** (0.081)	0.321 *** (0.081)	0.300 ** (0.080)	0.320 *** (0.080)	0.420 *** (0.100)
Government worker	0.553 ** (0.062)	0.560 *** (0.060)	0.651 *** (0.061)	0.560 *** (0.060)	0.662 * (0.063)
R2/Pseudo R2	0.047	0.047	0.048	0.047	0.050

Note: *** *p* < 0.01. ** *p* < 0.05. * *p* < 0.10. Robust standard errors appear in parentheses.

**Table 3 ijerph-20-02856-t003:** Results of empirical analysis based on heterogeneity.

Variable	Model (1)	Model (2)	Model (3)	Model (4)	Model (5)	Model (6)	Model (7)	Model (8)	Model (9)	Model (10)
	Group (1)	Group (2)	Group (1)	Group (2)	Group (1)	Group (2)	Group (1)	Group (2)	Group (1)	Group (2)
Adoption of Agricultural green production	0.195 *** (0.084)	0.337 *** (0.097)	—	—	—	—	—	—	—	—
Comprehensiveness	—	—	0.107 *** (0.073)	0.234 *** (0.081)	—	—	—	—	—	—
Water-saving irrigation technology	—	—		—	0.300 * (0.090)	0.396 *** (0.101)	—	—		—
Green pest control technology	—	—	—	—	—	—	0.151 ** (0.085)	0.223 *** (0.091)	—	—
Organic fertilizer substitution technology	—	—	—	—	—		—		0.232 *** (0.091)	0.315 ** (0.094)
Control variables	YES	YES	YES	YES	YES	YES	YES	YES	YES	YES
R2/Pseudo R2	0.042	0.042	0.043	0.042	0.051	0.052	0.042	0.050	0.044	0.050

Note: *** *p* < 0.01. ** *p* < 0.05. * *p* < 0.10. Robust standard errors appear in parentheses. Control variable estimation results are omitted but can be shared upon request.

**Table 4 ijerph-20-02856-t004:** Estimation results of the mediation effect of adopting agricultural green production on farmers’ happiness.

Variable	Model (1)	Model (2)	Model (3)	Model (4)	Model (5)	Model (6)	Model (7)	Model (8)	Model (9)	Model (10)
	Happiness	Increasing Absolute Income	Increasing Relative Income	Mitigating Agricultural Pollution	Elevating Social Status	Happiness				
Adoption of agricultural green production	0.455 *** (0.099)	0.681 *** (0.104)	0.463 *** (0.101)	0.360 ** (0.098)	0.679 *** (0.098)	0.342 *** (0.114)	0.213 ** (0.109)	0.199 *** (0.100)	0.220 ** (0.104)	0.233 * (0.118)
Increasing absolute income	—	—	—	—	—	0.077 ** (0.037)	—	—	—	0.075 * (0.044)
Increasing relative income	—	—		—	—	—	0.203 *** (0.037)	—		0.198 *** (0.043)
Mitigating agricultural pollution	—	—	—	—	—	—	—	0.132 *** (0.028)	—	0.127 *** (0.028)
Elevating Social status	—	—	—	—	—		—		0.287 *** (0.038)	0.262 *** (0.039)
Control variables	YES	YES	YES	YES	YES	YES	YES	YES	YES	YES
R2/Pseudo R2	0.119	0.187	0.163	0.066	0.113	0.121	0.130	0.127	0.139	0.155

Note: *** *p* < 0.01. ** *p* < 0.05. * *p* < 0.10. Robust standard errors appear in parentheses. Control variable estimation results are omitted but can be shared upon request.

**Table 5 ijerph-20-02856-t005:** Robustness testing results after replacing the explained variable assignment.

Variables	Model (1)	Model (2)	Model (3)	Model (4)	Model (5)	Model (6)	Model (7)	Model (8)	Model (9)	Model (10)
	Understate Happiness	Overstate Happiness	Understate Happiness	Overstate Happiness	Understate Happiness	Overstate Happiness	Understate Happiness	Overstate Happiness	Understate Happiness	Overstate Happiness
Adoption of Agricultural green production	0.201 *** (0.084)	0.496 *** (0.097)	—	—	—	—	—	—	—	—
Comprehensiveness	—	—	0.222 *** (0.085)	0.379 *** (0.090)	—	—	—	—	—	—
Water-saving irrigation technology	—	—		—	0.352 * (0.085)	0.410 *** (0.101)	—	—		—
Green pest control technology	—	—	—	—	—	—	0.354 ** (0.092)	0.505 *** (0.102)	—	—
Organic fertilizer substitution technology	—	—	—	—	—		—		0.231 *** (0.085)	0.433 ** (0.093)
Control variables	YES	YES	YES	YES	YES	YES	YES	YES	YES	YES
R2/Pseudo R2	0.037	0.037	0.038	0.037	0.039	0.039	0.037	0.037	0.036	0.036

Note: *** *p* < 0.01. ** *p* < 0.05. * *p* < 0.10. Robust standard errors appear in parentheses. Control variable estimation results are omitted but can be shared upon request.

**Table 6 ijerph-20-02856-t006:** Robustness testing results by confidence interval analysis.

Variables	Effect		95% Confidence Interval		Inspection Results
	Coefficient	Standard Error	Lower Limit	Upper Limit	
Adoption of agricultural green production	0.077	0.019	0.036	0.122	significant
Comprehensiveness	0.065	0.012	0.038	0.089	significant
Water-saving irrigation technology	0.084	0.023	0.032	0.136	significant
Green pest control technology	0.029	0.009	0.009	0.049	significant
Organic fertilizer substitution technology	0.080	0.019	0.037	0.123	significant

Note: The bias-corrected nonparametric percentile bootstrap method was used here with 5000 replicates.

**Table 7 ijerph-20-02856-t007:** Robustness testing results by partial samples.

Variable	Model (1)	Model (2)	Model (3)	Model (4)	Model (5)	Model (6)	Model (7)	Model (8)	Model (9)	Model (10)
	Happiness	Increasing Absolute Income	Increasing Relative Income	Mitigating Agricultural Pollution	Elevating Social Status	Happiness				
Adoption of agricultural green production	0.557 *** (0.114)	0.314 ** (0.109)	0.533 *** (0.100)	0.339 *** (0.104)	0.208 * (0.118)	0.386 *** (0.114)	0.337 ** (0.109)	0.331 *** (0.100)	0.326 *** (0.104)	0.208 ** (0.118)
Increasing absolute income	—	—	—	—	—	0.356 *** (0.099)	—	—	—	0.301 *** (0.098)
Increasing relative income	—	—		—	—	—	0.377 ** (0.111)	—	—	0.339 *** (0.098)
Mitigating agricultural pollution	—	—	—	—	—	—	—	0.230 *** (0.085)	—	0.127 ** (0.080)
Elevating Social status	—	—	—	—	—	—	—	—	0.098 *** (0.050)	0.075 *** (0.038)
Control variables	YES	YES	YES	YES	YES	YES	YES	YES	YES	YES
R2/Pseudo R2	0.201	0.201	0.202	0.201	0.203	0.200	0.201	0.127	0.139	0.155

Note: *** *p* < 0.01. ** *p* < 0.05. * *p* < 0.10. Robust standard errors appear in parentheses. Control variable estimation results are omitted but can be shared upon request.

**Table 8 ijerph-20-02856-t008:** Balance test results of explanatory variables before and after matching.

Matching Method	Pseudo R2	LR	*p*	MeanBias	MedBias
Before matching	0.486	549.68	0.000	95.3	83.6
Nearest neighbor matching (k = 1)	0.120	196.91	0.000	26.7	19.5
Nearest neighbor matching (k = 4)	0.112	211.20	0.000	25.0	14.5
Kernel matching (0.06)	0.130	260.01	0.000	29.9	23.1
Kernel matching (0.10)	0.134	266.68	0.000	30.6	22.9

**Table 9 ijerph-20-02856-t009:** Treatment effects (ATT) using matching exercises.

Matching Method	Treat Group	Control Group	ATT	Standard Error	t
Before matching	3.626	3.116	0.510	0.058	8.73
Kernel matching (0.06)	3.625	3.138	0.487	0.178	2.75
Kernel matching (0.10)	3.625	3.153	0.472	0.174	2.73
Nearest neighbor matching (k = 1)	3.568	3.081	0.487	0.205	2.38
Nearest neighbor matching (k = 4)	3.568	3.143	0.425	0.186	2.29

## Data Availability

The data used in this study contain sensitive information about the study participants, and they did not provide consent for public data sharing. A minimal data set could be shared upon request by a qualified academic investigator for the sole purpose of replicating the present study.

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
