# Peer review of "From Agricultural Green Production to Farmers’ Happiness: A Case Study of Kiwi Growers in China"

_ijerph, 2023, doi:10.3390/ijerph20042856_

Round 1
Reviewer 1 Report
Please, see the attached document

Reviewer 2 Report
Authors have done an excellent job by putting this manuscript together.
Reviewer 3 Report
The research behind this paper seems really important and results are based on more than 1,100 answers in a survey. That is certainly impressive and implies a great effort. The way the research is envisioned, looking for a relationship between happiness, as intangible and unmeasurable it is, and green agriculture is refreshing and indeed interesting but reviewing it from a scientific point of view there are several assumptions and comments that have to be checked and corrected.
First, introduction, concept definition and literature review include many statements without no reference at all. Any serious academic paper has to provide information about the sources of the pieces of information that is included in the document. In addition to that some references don't add up. For example, non of the cited references in line 111 seems to distinguish between extrinsic and intrinsic factors regarding happiness. Listing the marital status or the frequency of sexual intercourse in relation to happiness in a paper related to green agriculture seems quite out of place. The same is found when explaining the destination country regarding migrant's happiness, sure it is important but not relevant here.
Regarding the design of the proposal, it seems risky to assume that engagement in agricultural green production is achieved by just the adoption of one of the studied technologies, regardless impact, moment of time or any other consideration. Nevertheless, no reference or previous work is provided to back this assumption made by authors.
Figure 1 should be placed later in the paper, when all relevant factors have already been explained in the text.
Some words seem to be missing in lines 260-262 because it is really difficult to understand what authors intend to say.
Labels in Figure 3 are misplaced because percentages do not correspond to the shares in the graph.
The statistical methods applied seem quite appropiate although from my point of view the main weakness is the measurement of the variable "happiness". It is recognised by the authors themselves that they use a simple method (asking the farmers directly), but this value is so subjective, has no way of comparison to a situation where the same subject was not engaged in green activities, so you can check using panel data if engaging in these activities leads to a happier situation. I am sorry I fail to see scientific relevance in the findings of the relation between the personal perception in a given moment and the engagement in sustainable activities
Round 2
Reviewer 1 Report
The authors have incorporated all my comments and suggestions in the paper. Therefore, I recommend accepting it in its present form.
Reviewer 3 Report
Authors have significantly improved the document and it is ready to be published from my humble point of view